# Performance of Five Metagenomic Classifiers for Virus Pathogen Detection Using Respiratory Samples from a Clinical Cohort

**DOI:** 10.3390/pathogens11030340

**Published:** 2022-03-11

**Authors:** Ellen C. Carbo, Igor A. Sidorov, Anneloes L. van Rijn-Klink, Nikos Pappas, Sander van Boheemen, Hailiang Mei, Pieter S. Hiemstra, Tomas M. Eagan, Eric C. J. Claas, Aloys C. M. Kroes, Jutte J. C. de Vries

**Affiliations:** 1Department of Medical Microbiology, Leiden University Medical Center, 2333 ZA Leiden, The Netherlands; i.sidorov@lumc.nl (I.A.S.); a.vanrijn@hagaziekenhuis.nl (A.L.v.R.-K.); s.vanboheemen@erasmusmc.nl (S.v.B.); e.c.j.claas@lumc.nl (E.C.J.C.); a.c.m.kroes@lumc.nl (A.C.M.K.); jjcdevries@lumc.nl (J.J.C.d.V.); 2Sequencing Analysis Support Core, Department of Biomedical Data Sciences, Leiden University Medical Center, 2333 ZA Leiden, The Netherlands; n.pappas@uu.nl (N.P.); h.mei@lumc.nl (H.M.); 3Theoretical Biology and Bioinformatics, Department of Biology, Science for Life, Utrecht University, 3584 CH Utrecht, The Netherlands; 4Department of Viroscience, Erasmus Medical Center, 3015 GD Rotterdam, The Netherlands; 5Department of Pulmonology, Leiden University Medical Center, 2333 ZA Leiden, The Netherlands; p.s.hiemstra@lumc.nl; 6Department of Thoracic Medicine, Haukeland University Hospital, 5021 Bergen, Norway; tomas.eagan@med.uib.no

**Keywords:** viral metagenomics, bioinformatics, pathogen detection, next-generation sequencing

## Abstract

Viral metagenomics is increasingly applied in clinical diagnostic settings for detection of pathogenic viruses. While several benchmarking studies have been published on the use of metagenomic classifiers for abundance and diversity profiling of bacterial populations, studies on the comparative performance of the classifiers for virus pathogen detection are scarce. In this study, metagenomic data sets (*n* = 88) from a clinical cohort of patients with respiratory complaints were used for comparison of the performance of five taxonomic classifiers: Centrifuge, Clark, Kaiju, Kraken2, and Genome Detective. A total of 1144 positive and negative PCR results for a total of 13 respiratory viruses were used as gold standard. Sensitivity and specificity of these classifiers ranged from 83 to 100% and 90 to 99%, respectively, and was dependent on the classification level and data pre-processing. Exclusion of human reads generally resulted in increased specificity. Normalization of read counts for genome length resulted in a minor effect on overall performance, however it negatively affected the detection of targets with read counts around detection level. Correlation of sequence read counts with PCR Ct-values varied per classifier, data pre-processing (R^2^ range 15.1–63.4%), and per virus, with outliers up to 3 log_10_ reads magnitude beyond the predicted read count for viruses with high sequence diversity. In this benchmarking study, sensitivity and specificity were within the ranges of use for diagnostic practice when the cut-off for defining a positive result was considered per classifier.

## 1. Introduction

In the era of next-generation sequencing (NGS), clinical metagenomics, the analysis of all microbial genetic material in clinical samples, is being introduced in diagnostic laboratories and revolutionizing the diagnostics of infectious diseases [1,2,3,4]. As opposed to running a series of pathogen targeted diagnostic PCR assays to identify suspected pathogens, one single metagenomic run enables the detection of all potential pathogens in a clinical sample [5,6]. The use of this method, also known as shotgun high-throughput sequencing, has resulted in the detection of several pathogens missed by current routine diagnostic procedures [1,7]. For a large part the clinical application of metagenomic sequencing for pathogen detection has focused on patients with encephalitis [1,8,9,10,11,12]. However, patients with clinical syndromes suspected from an infectious disease but with negative conventional test results are increasingly considered as candidates for metagenomic testing. With sequencing costs decreasing and the significance of detection of unexpected, novel viruses being underscored by the currently pandemic SARS-CoV-2 [13], metagenomics is increasingly moving towards implementation in diagnostic laboratories.

Performance testing is typically part of the implementation procedure in diagnostic laboratories to ensure the quality of diagnostic test results. Accurate bioinformatic identification of viral pathogens depends on both the classification algorithm and the database [14,15,16]. Metagenomic sequencing in the past has been mainly oriented at profiling of bacterial genomes in the context of microbiome comparisons in research settings, and most bioinformatic tools currently available have been designed for that specific purpose [17,18]. Some of the previously bacterial oriented classifiers are now being used for other domains, including viruses. However, viral metagenomics for pathogen detection has specific challenges such as the low abundancy of viral sequences for some targets, and incomplete or inaccurate reference sequences. The high diversity of viral sequences due to the high mutation rate of RNA viruses further complicates accurate detection and identification [19]. While the number of benchmarking studies published on the use of metagenomic classifiers for bacterial abundancy profiling is increasing, studies on the performance of classifiers for virus pathogen detection remain scarce. Publications on the performance of the computational analysis of viral metagenomics are usually limited to in silico analysis of artificial sequence data [14,20,21] or mock samples [22,23]. Though both sensitivity and specificity can be deduced when using simulated datasets, they usually do not represent the complexity of data sets from clinical samples which typically contain sequences from wet lab reagents that have been referred to as the ‘kitome’ [22,24,25]. These factors can affect the sensitivity and specificity of the overall procedure and may result in incorrect diagnoses. In contrast, performance studies that use real-world samples are usually hindered by the huge number of negative metagenomic findings in the absence of gold standard results for validation. Therefore, the performance parameters typically reported are recall (sensitivity), precision (positive predictive value), and F1 (the harmonic mean of recall and precision); while specificity is usually not assessed because negative findings by metagenomics are poorly defined.

Here, we perform a comparison of five taxonomic classifiers: Centrifuge [26], Clark [18], Kaiju [27], Kraken 2 [28], and Genome Detective [29]. The classifiers were tested using metagenomic shotgun sequencing data obtained from a cohort of chronic obstructive pulmonary disease patients (COPD) with a clinical exacerbation and therefore suspected of a respiratory infection. For these samples, 1144 PCR test results were used as gold standard to infer both sensitivity and specificity of the classifiers. For each classifier, we present appropriate benchmark scores for virus classification in the diagnostic setting.

## 2. Materials and Methods

### 2.1. Clinical Samples and PCR Results

Clinical respiratory samples were used to obtain metagenomic data sets. In total 88 nasal washings were taken from 63 patients with COPD suspected for respiratory infection as previously described [30]. Each sample was tested using a respiratory PCR panel resulting in 1144 real-time positive and negative PCR results for 13 viral respiratory targets as previously described [30]. The respiratory viruses addressed by this respiratory panel and cohort prevalence are shown in Table 1.

### 2.2. Metagenomic Next-Generation Sequencing (mNGS)

The metagenomic datasets used for comparison were generated as described before [30]. In short, clinical samples were spiked with equine arteritis virus (EAV) and phocine herpesvirus 1 (PhHV-1), as internal positive controls for RNA and DNA detection per sample, throughout the entire workflow. Negative and positive washings were used as respectively environmental and positive run controls. Subsequently, extraction of nucleic acids was performed using the Magnapure 96 DNA and Viral NA Small volume extraction kit on the MagnaPure 96 system (Roche, Basel, Switzerland). Library preparation was performed utilizing the NEBNext Ultra II Directional RNA Library prep kit for Illumina (New England Biolabs, Ipswich, MA, USA) using single, unique adaptors and a protocol optimized for processing RNA and DNA simultaneously in a single tube [25]. Sequencing was performed on an Illumina NextSeq 500 sequencing system (Illumina, San Diego, CA, USA) at GenomeScan BV (Leiden, The Netherlands), obtaining approximately 10 million 150 bp paired-end reads per sample.

### 2.3. Pre-Processing of Data

To exclude variability based on pre-processing procedures, the identical procedure was followed prior to analysis of the sequence data by all classifiers in the current comparison. Illumina 150 bp paired-end sequence reads were demultiplexed by standard Illumina software followed by trimming, adapter clipping, and filtering of low-complexity reads using Trimmomatic [v. 0.36] [31]. This was performed for all classifiers, regardless of quality filtering options that have been previously used in combination with specific classifiers in literature. Human reads were excluded after mapping them to the human genome GRCh38 [32] using Bowtie2 with standard settings [33]. Unmapped reads were used for further analysis for the classification tests excluding human reads.

### 2.4. Metagenomic Classifiers

Bioinformatic metagenomics tools designed for taxonomic classification were selected for benchmarking based on the following criteria: applicable for viral metagenomics for pathogen detection; available either as download or webserver; and it is either widely used or showed potential of diagnostics implementation in the future. Some tools considered were excluded due to lack of support or details on how to use the tool, or non-functioning webservers. An overview of characteristics of the selected classifiers can be found in Table 2.

### 2.5. Reference Database

For comparison of classification performance, a single database was used as starting point for the classifiers Centrifuge, Clark, Kaiju, and Kraken 2: viral genomes from NCBI/RefSeq [34] (downloaded on 27 December 2020). Genome Detective was used as a service, and it uses its own database that was generated on 3 March 2020 (version 1.130) by Genome Detective.

### 2.6. Metagenomic Classifiers and Characteristics

#### 2.6.1. Centrifuge

Classification with Centrifuge (version 1.0.4) [26] is based on exact matches of at least 22 base pair nucleotide sequences with the reference index, using *k*-mers of user-defined length. Centrifuge by default allows five classification labels per sequence read. For a realistic comparison, in the current study, this setting was adapted to maximum one label per sequence (the lowest common ancestor) to mimic results of Kraken2 and other classifiers where only one label per sequence read is given. Preceding classification, Centrifuge builds small reference indexes based on adapted versions of the Burrows–Wheeler transform (BWT) [35] and the Ferragina–Manzini (FM) index [36] resulting in a compressed index of only unique genomic sequences.

#### 2.6.2. Clark

Clark (version 1.2.6.1) [18] is a taxonomic classifier based on reduced *k*-mers using nucleotide-level classification. It uses a compressed index database containing unique target specific k-spectrum of target sequences. For the current comparison the default execution mode was used.

#### 2.6.3. Kaiju

Kaiju (version 1.7.3) [27] is a taxonomic classifier that assigns sequence reads using amino acid-level classification. Sequence reads are translated into six possible open reading frames and split into fragments according to the detected stop codons. Classification with Kaiju can be performed using two settings, both based on an adjusted backward alignment search algorithm of BWT [35]. For the current comparison study, the greedy mode was used providing high sensitivity because it allows up to five mismatches to further increase the highest scoring matches. In this mode Kaiju assesses six possible ORF’s using the amino acid scores of Blosum62 [37] to obtain the highest scoring match.

#### 2.6.4. Kraken 2

Kraken 2 (version 2.0.8-beta) [28] is a classifier designed to improve the large memory requirements of the former version of Kraken [17], resulting in a reduction of in general 85% of the size of the index database. Kraken 2 uses a probabilistic, compact hash table to map minimizers to the lowest common ancestors (LCA), and stores only minimizers from the reference sequence library in its index reference [28].

#### 2.6.5. Genome Detective

Genome Detective [29] is a commercially available bioinformatic pipeline that includes the entire workflow from automated quality control, de novo assembly of reads and classification of viruses. After automated adapter trimming and filtering low-quality reads using Trimmomatic [31], viral reads are selected based on Diamond [38] protein alignment using as reference protein sequences from Swissprot Uniref 90 [39]. Viral reads are sorted in buckets, after which all sequences in one bucket are de novo assembled into contigs using SPAdes [40] or metaSPAdes [41]. Subsequently, contigs are processed by BLASTx and BLASTn [42] against databases containing NCBI Refseq [34] sequences and some additional virus sequences. Potential hits represented by the contigs are assigned to individual species using the Advanced Genome Aligner [43], and coverage the viral genomes is calculated. For analysis using Genome Detective sequence reads were first pre-processed with Trimmomatic [31] manually, similar for other tools (see Pre-processing of data), prior to automated filtering by the Genome Detective pipeline.

### 2.7. Performance, Statistical Analysis, and ROC

Sensitivity and specificity were calculated for the classifiers based on the application of PCRs (designed for detection of 13 targets) for 88 samples with 24 PCR positive and 1120 PCR negative results. Receiver Operating Characteristic (ROC) curves were generated for results of classification at species, genus, and family levels, by varying the number of sequence-read counts used as cut-off for defining a positive result (resolution: 1000 steps from one read to the maximum number of sequence reads for each PCR target per sample). Area under the curve (AUC), the ROC distance to the closest error-free point (0,1, informedness) curve, positive and negative predictive values were calculated. Furthermore, correlation (R^2^) of sequence read counts with PCR cycle threshold (Ct) value were analyzed.

## 3. Results

### 3.1. Performance: Sensitivity, Specificity, and ROC

The performance of the selected taxonomic classifiers Centrifuge, Clark, Kaiju, Kraken 2, and Genome Detective for metagenomic virus pathogen detection was assessed using datasets from 88 respiratory samples with 24 positive and 1120 negative PCR results available as gold standard. To exclude variability based on different default databases provided with the classifiers, a single database of reference genome sequences was used in combination with a standardized dataset for all classifiers. Raw NGS reads were filtered and classified, both prior and after the exclusion of human sequence reads, and after exclusion of human reads combined with normalization of reads based on the target viral genome length. ROC curves are shown for all classifiers, for assignments at species, genus and family level for the NGS data in Figure 1, and Appendix A. Detection parameters (ROC distance to the upper left corner of the graph, sensitivity and selectivity, and AUC) at three taxonomic levels calculated for the NGS data, before and after exclusion of human reads, with or without normalization of assigned reads by corresponding genome sequence lengths are additionally shown in Figure 2. Overall, sensitivity, specificity, and AUC ranged from 83 to 100%, 90 to 99%, and 91 to 98%, respectively, and varied per level of taxonomic classification, per classifier, and with the exclusion of human reads prior to classification. Classification at species and genus levels tended to result in lower sensitivity and higher ROC distances, but higher selectivity when compared with family level classification, for most of the classifiers evaluated. Extraction of human sequence reads prior to classification resulted in comparable sensitivity at all levels of assignment for all classifiers except CLARK for which sensitivity plummeted at species and genus levels. Selectivity was mainly increased after extraction of human reads, for classification at all levels, except for Kaiju and Kraken2, for which decreased selectivity values at family level were observed. Extraction of human reads reduced the differences in selectivity between the classifiers that were observed at genus and family level prior to extraction. The ROC distances were overall smallest, and the AUC highest, when using amino-acid based classifier Kaiju, the latter at species and family levels and was comparable with Kraken2 at genus level. Normalization of assigned read counts by corresponding genome length resulted in minor changes in performance when considering 1 read as the threshold for defining positive results. Sensitivity was dramatically reduced to 13–33% at species level after read normalization when a threshold of 10 reads was applied, while sensitivity was 75–88% without read normalization in combination with a threshold of 10 reads, (Appendix A). This indicates that normalization of reads can negatively affect the detection of targets with read counts around detection level.

Overall, Kaiju outperformed all classifiers when ROC distance, AUC, and sensitivity were considered, but had consistently lower selectivity values than Centrifuge and Genome Detective.

In this patient cohort, with an incidence of 21% (24/88 samples) of respiratory viruses, the positive and negative predictive values at species levels were 42–67% and 99–100%, respectively (see Appendix A).

### 3.2. Correlation Read Counts and Ct-Values

The correlation between sequence read counts at Ct-value for the corresponding PCR target viruses for all classifiers is shown in Figure 3 and Appendix A. Correlation (R2, %), linear regression slope and intercept varied per virus species, per taxonomic classifier, and was dependent on the extraction of human reads. Correlation ranged from 15.1% for CLARK (no exclusion of human reads, species level) to 62.7% for Kaiju-based classification at species level (after exclusion of human reads with normalization of assigned reads by corresponding genome sequence lengths). The most consistent results (when comparing R^2^ prior and after human reads exclusion, and after normalization) were demonstrated by Kaiju and Genome Detective with overall outperformance of Kaiju classifier at all classification levels (61.8–62.7% versus 42.3–43.9% for Centrifuge). Reads assigned to rhinoviruses were most common outliers in relation to Ct-value and varied up to 3 log_10_ reads difference from the predicted read count (LR), possibly resulting from their high divergence within species. This was in contrast to read counts of other viruses (for example influenza viruses), which were closer to the predicted correlation line. Extraction of human sequence reads resulted in an increase in R^2^ for CLARK classifier at species and family level, a decrease for Centrifuge and Kraken at all levels, and resulted in minor changes for amino acid-based classifiers Genome Detective and Kaiju at all levels. Decrease in absolute or relative number of total reads after pre-processing (extraction of human reads in combination with normalization of assigned reads by corresponding genome lengths) led to a decrease in intercept values for all classifiers.

These data support that a more accurate taxonomic classification assists semi-quantitative performance of metagenomic classification tools.

## 4. Discussion

In this study, we compared the performance of five taxonomic classification tools for virus pathogen detection, using datasets from well-characterized clinical samples. In contrast to previously reported comparisons with datasets from real samples, both sensitivity and specificity could be assessed using a unique set of 1144 PCR results as gold standard. A uniform database was created to exclude variability based on differences in availability of genomes in databases provided with the classifiers. In general, sensitivity and specificity were within ranges applicable to diagnostic practice. Exclusion of human reads generally resulted in increased specificity. Normalization of read counts for genome length negatively affected the detection of targets with read counts around detection level. The correlation of sequence read counts with PCR Ct-values was highest for viruses with relatively lower sequence diversity.

Previous studies have benchmarked metagenomic profilers, mainly for the use of bacterial profiling and DNA-to-DNA and DNA-to-protein classification methods were among the best-scoring methods in comparison with DNA-to-marker (16S) methods [22,27,44,45,46,47,48]. In a study with simulated bacterial datasets comparing the performance of CLARK, Kraken and Kaiju, sensitivity and precision were 75% and 95% and decreased when a lower number of reference genomes was available for the specific target [27]. As the same reference database was used by all classifiers in this study, the only determining factors would be the index database built from the reference database and the classification algorithm. DNA-to-DNA methods have been applied in hundreds of published microbiome studies (e.g., Kraken: 1438 citations; Kraken 2: 204 citations, by March 2021, according to their official websites [48]). Centrifuge was designed as a follow-up of Kraken with enhanced features, though misclassifications have also been reported in a comparison with simulated datasets [22]. DNA-to-protein methods are generally more sensitive to novel and highly variable sequences due to lower mutation rates of amino acid compared to nucleotide sequences [22,27] as was seen in our study when classifying rhinoviruses by Kaiju. The difference was especially visible in genera with limited availability of genomes in reference databases [27].

Misclassification of human genomic sequence reads has been reported for most DNA classifiers [22]. Protein-based classifiers had higher misclassification ranges of human genome sequences (up to 15%), partially due to the larger number of target sequences in their default databases [22]. Inclusion of the human genome in the reference database, which is by default the case for Centrifuge and KrakenUniq [49] reduced the rate of misclassification to negligible [22]. This finding is supported in our study, as exclusion of human sequence reads prior to classification reduced misclassifications for all classifiers. In general, reduction of false-positive hits can be achieved by assembly of sequences (for example, by Genome Detective), thus reducing the number of hits based on short nucleotide sequences used by *k*-mer based methods. Inclusion of genome coverage of mapped reads, as adopted by Genome Detective and KrakenUniq [49], also can reduce false-positive hits.

One of the strengths of this study, the use of one single wet lab and sequencing procedure, in order to enable comparison of the bioinformatic analyses, is also a limitation of the study. The sensitivity and specificity results will likely vary when the classifiers are used in combination with a different wet lab methodology. Therefore, no conclusions can be drawn on the absolute numbers, sensitivity and specificity, of other workflows that include the classifiers, since every step in the entire workflow can influence the overall performance.

To our knowledge, a limited number of studies on the benchmarking of tools for viral metagenomics for pathogen detection have been published. In a Switzerland-wide ring trial based on spiked plasma samples, median F1 scores ranged from 70 to 100% for the different pipelines, though since the entire workflow was analyzed, and thus no conclusions on specific classifiers could be drawn [15]. A series of tools and programs were analyzed in a COMPARE virus proficiency test using a single in silico dataset [14]. For Kraken discrepant classification results that were observed, this was likely due to differences in the databases used by the participants. A recent European benchmark of 13 bioinformatic pipelines currently in use for metagenomic virus diagnostics used datasets from clinical samples [16] analyses using Centrifuge and Genome Detective software resulted in sensitivities of 93% and 87%, respectively.

In conclusion, sensitivity and specificity of the classifiers evaluated in this study was within the ranges that may be applied in clinical diagnostic settings. Performance testing for viral metagenomics for pathogen detection is intrinsically different from benchmarking of bacterial profiling and should incorporate parameters that are inherent to clinical diagnostic use such as specificity calculations, sensitivity for divergent viruses and variants, and importantly, a determined cut-off for defining a positive result for each workflow. Taking these factors into account during validation and implementation of viral metagenomics for pathogen detection contributes to optimal performance and applicability in clinical diagnostic settings.

## Figures and Tables

**Figure 1 pathogens-11-00340-f001:**
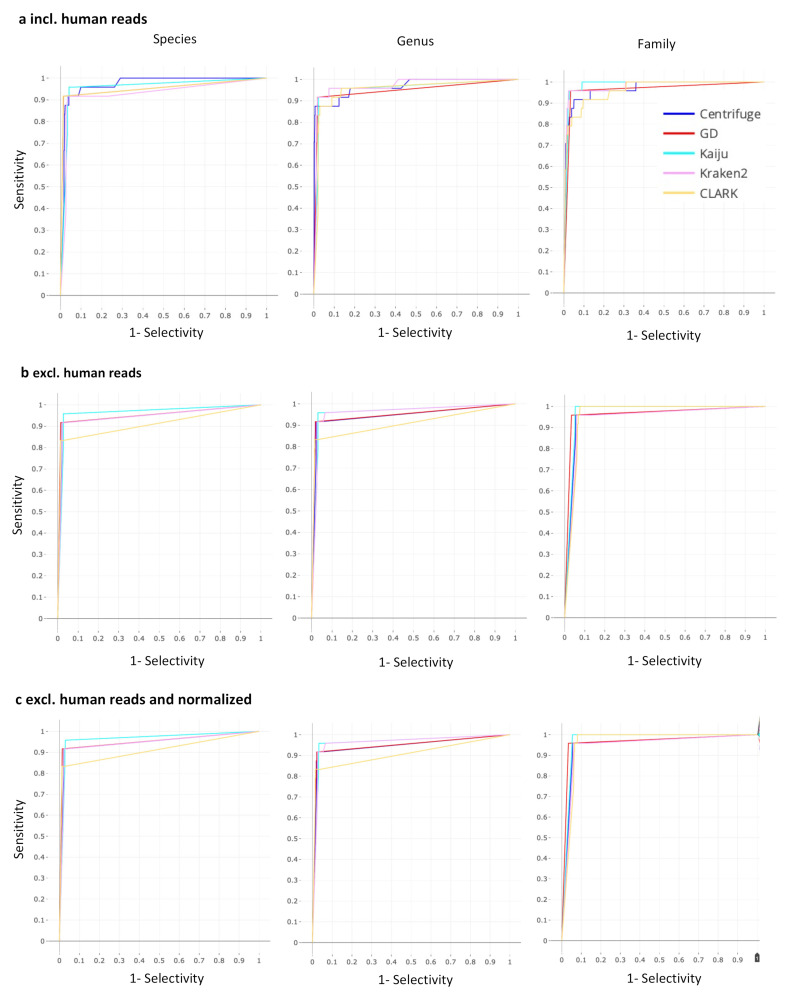
ROC curves calculated based on reads of taxonomic assignment at three. taxonomic levels (species, genus, and family) by the five classifiers, based on PCR-targets, (**a**), without extraction of human reads and (**b**), after extraction of human reads, (**c**), after extraction of human reads and normalization of reads by corresponding genome lengths (resolution of 1000 steps from one read to the maximum number of sequence reads for each PCR target per sample).

**Figure 2 pathogens-11-00340-f002:**
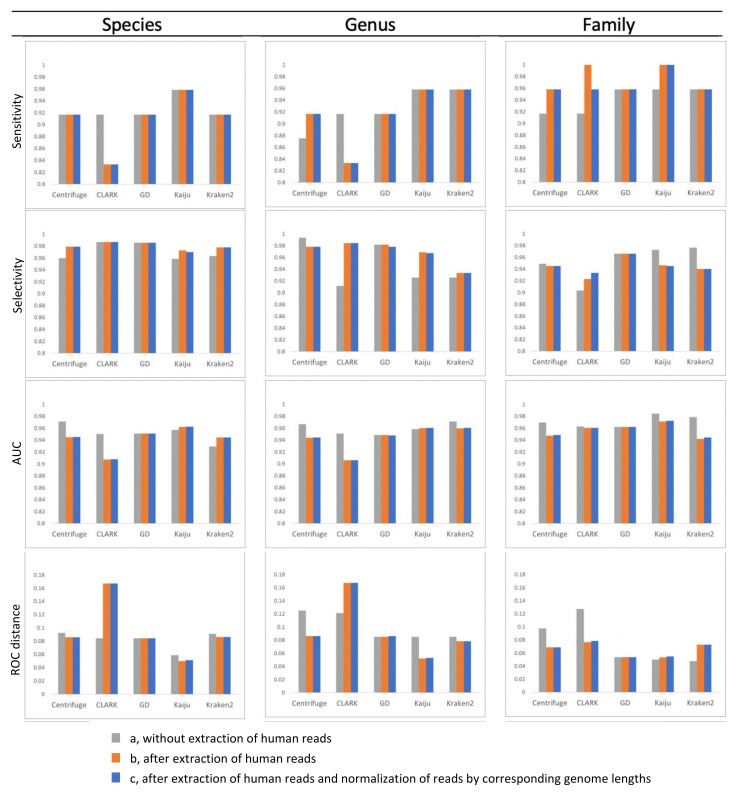
Sensitivity, selectivity, AUC, and ROC distance calculated based on assignment at three taxonomic levels (species, genus, and family) by the five classifiers for three types of pre-processing of the NGS datasets, a, without extraction of human reads and b, after extraction of human reads, c, after extraction of human reads and normalization of reads by corresponding genome lengths.

**Figure 3 pathogens-11-00340-f003:**
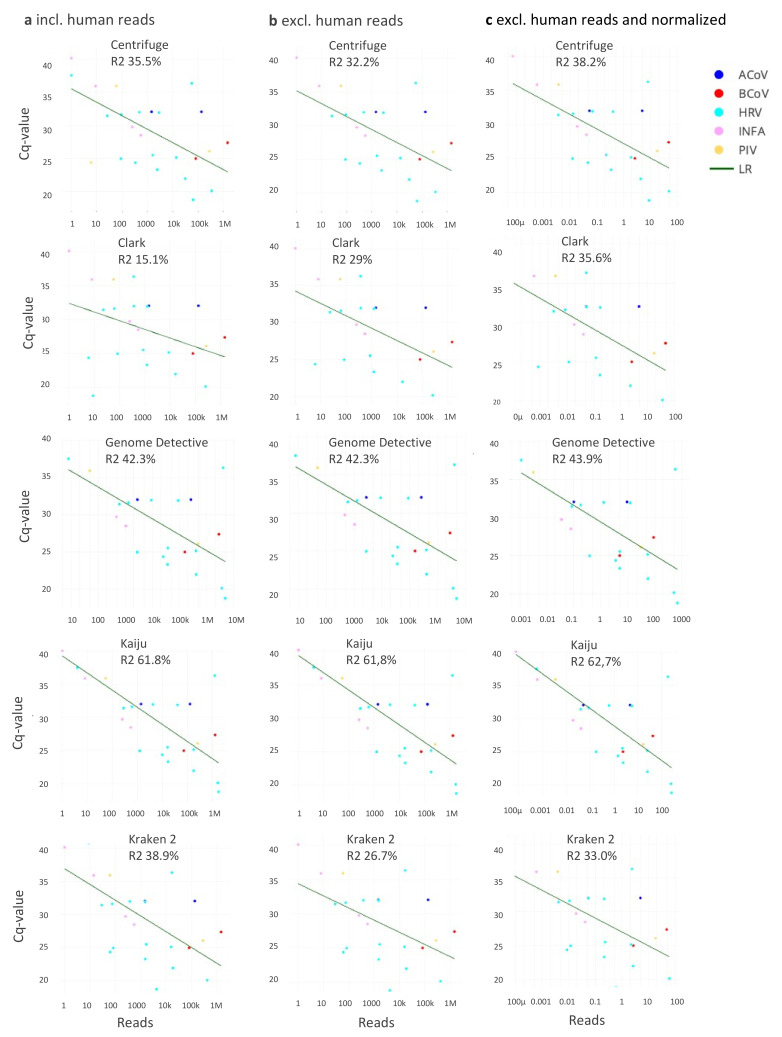
Correlation between the number of sequence reads assigned (species level) and Ct-values of virus-specific PCRs, for the five taxonomic classifiers evaluated, (**a**), without extraction of human reads and (**b**), after extraction of human reads, (**c**), after normalization of reads by corresponding genome lengths.

**Table 1 pathogens-11-00340-t001:** Overview of respiratory PCR panel targets and their test results.

PCR	Family	Genus	Species	Alternative Naming	# PCR Positive Samples	# PCR Negative Samples	PCR Ct-Values
Target Viruses							(Range)
HRV	*Picorna-viridae*	*Enterovirus*	*Rhinovirus A, B, C, Enterovirus D*		14	74	19–38
PIV1, PIV3	*Paramyxo-viridae*	*Respiro-virus*	*Human respirovirus 1*	*Human parainfluenza virus 1*	-	88	-
			*Human respirovirus 3*	*Human parainfluenza virus 3*	2	86	26–36
PIV2, PIV4	*Paramyxo-viridae*	*Ortho-rubulavirus*	*Human orthorubulavirus 2*	*Human parainfluenza virus 2*	-	88	-
			*Human orthorubulavirus 4*	*Human parainfluenza virus 4*	1	87	24
INF	*Orthomyxoviridae*	*Alphainfluenzavirus*	*Influenza A virus*		3	85	29–36
			*Influenza B virus*		-	88	-
ACoV	*Corona-viridae*	*Alpha-coronavirus*	*Human coronavirus NL63*		2	86	32
			*Human coronavirus 229E*		-	88	-
BCoV	*Corona-viridae*	*Betacoronavirus*	*Human coronavirus HKU1, Betacoronavirus 1; Human coronavirus OC43*		2	86	27
HMPV	*Pneumo-viridae*	*Metapneumovirus*	*Human metapneumo-virus*		-	88	-
RSV	*Pneumo-viridae*	*Orthopneumovirus*	*Human orthopneumo-virus*		-	88	-
Total			Total PCR results: 1144 (13 targets tested in 88 samples)		24	1120	19–38

**Table 2 pathogens-11-00340-t002:** Overview of characteristics of the classifiers evaluated.

	Centrifuge [26]	Clark [18]	Kaiju [27]	Kraken 2 [28]	Genome Detective [29]
License	Open source	Open source	Open source	Open source	Commercial/free to use web application
Version	1.0.4	1.2.6.1	1.7.3	2.0.8-beta	1.126
Sequencing technology compatibility	Short/long reads	Short/long reads	Short/long reads	Short/long reads	Short reads (long reads experimentally)
Pre-processing	No	No	No	No	Yes
Type of alignment	NT	NT	AA	NT	NT/AA including de novo assembly
Algorithm characteristics	Exact matches of 22 bp with target with default five labels per sequence, LCA optional	Exact matches of 31 bp with target with highest number of hits	Maximum exact matches (MEM) of AA, up to five mismatched optional *. LCA in case of multiple hits	Exact matches of 35 bp. LCA in case of multiple hits	Combined results of NT and AA hits based on scoring. LCA in case of multiple hits
Database (compression)	Compressed index NT database of only unique sequences	Compressed index NT database of only unique sequences	No compression, AA database	Compressed index NT database	No compression, viral subset of Swiss-Prot UniRef90 protein database

NT; nucleotide, AA; amino acid; LCA, lowest common ancestor. * Greedy-5 mode was used in the current study.

## Data Availability

NGS data used in this study have been submitted (after removal of human reads) to the NCBI’s Sequence Read Archive (http://www.ncbi.nlm.nih.gov; accession number SRX6713943-SRX6714030).

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
