# Peer review of "Performance of Five Metagenomic Classifiers for Virus Pathogen Detection Using Respiratory Samples from a Clinical Cohort"

_pathogens, 2022, doi:10.3390/pathogens11030340_

Round 1
Reviewer 1 Report
The authors have done a comprehensive study on a well described sample set. The problem with this review is, that I cannot read a single figure as the description and scales are not readable. It is also not clear if the y-axis is comparable on all figures?
I am happy to read it again, once the figures are clear.
Author Response
We regret sincerely that the reviewer was not able to correctly see all figures. In our uploads figures had a good resolution, maybe resolution was lost when converted to the reviewer pdf. Regardless we would like to thank the reviewer for the comments. We have adapted the font size in all figures, though for comparison we would like all graphs on one page.
We hope it is more clear to the reviewer now.
Reviewer 2 Report
Please find my remarks below:
Line 51: [1-4] rather than [1]–[4]. Formats of other citations also seem to recquire correction.
Table 1.: Generic and specific names should be italicized. Also those of viruses. Moreover, many genera and species are typed with gaps (or lack them) in inappropriate positions.
Author Response
Line 51: [1-4] rather than [1]–[4]. Formats of other citations also seem to recquire correction.
Dear reviewer, we thank you for your time to look at our manuscript and for your remark on the citations. We corrected the format of the citations, so now its according to a suggested [1-4] format.
Table 1.: Generic and specific names should be italicized. Also those of viruses. Moreover, many genera and species are typed with gaps (or lack them) in inappropriate positions.
We have corrected this, generic and specific names are in italic in the revised version.
Reviewer 3 Report
The study performed Carbo et al., describes an interesting point of the metagenomic analysis, comparing different taxonomic classifiers which signficantly can influence the final result of the next generation sequencing. Depending on the used taxonomic method we can obtain different results which in fact influence all of our further analysis. I have the following doubts:
I did not find the complete pipeline for metagenomic analysis. What were the inital steps before the taxonomic classification. The pipelines varied in some parameters or not? Quality control, cleaning of the host sequences, adapters etc. All these steps can influence the metagenomic analyses.
How the authors led the analysis related to contaminants and viral artifacts? Viral contaminants, vectors, erroneus- classification? The authors observed contaminats? Used specific programs ?
The classfication was performed based on only on the generated viral reads or formed contigs? Depending on the Ct it was possible to assemble complete genomes? What coverage?
What types of controls were used in all runs? Non-template, negative, sequencing?
The discussion needs imporvement.
Author Response
We thank the reviewer for the very useful comments. We put all the adjustments in the manuscript and explain point by point in the reply/pdf document to the reviewer what we exactly changed.

Round 2
Reviewer 3 Report
The authors replied to all the questions. I think the manuscript is well organized now.